# Cost-Effectiveness Analysis of Hp and New Gastric Cancer Screening Scoring System for Screening and Prevention of Gastric Cancer

Peiyu Zheng [1] and Jinchun Liu [1,2,*]

[1] Graduate School, Shanxi Medical University, Taiyuan 030001, China
[2] Department of Gastroenterology, The First Hospital of Shanxi Medical University, Taiyuan 030001, China
* Correspondence: zxr610624@163.com; Tel.: +86-0354-6286909

**Abstract:** Gastric cancer is one of the most common gastrointestinal cancers. Early diagnosis can improve the 5-year survival rate. This study aimed to evaluate the cost-effectiveness of Helicobacter pylori (Hp) and a new gastric cancer screening scoring system (NGCS) in areas with a high incidence of gastric cancer. A decision-analytic Markov model was constructed based on the theory and method of cost-effectiveness analysis, which included three decisions: no screening, Hp screening, and NGCS screening. The uncertainty of each parameter in the model was determined using a one-way sensitivity analysis and probability sensitivity analysis. The results of the cost-effectiveness analysis revealed that the application of the NGCS had the highest cost-effectiveness, while the one-way sensitivity analysis revealed that the probability of intestinal metaplasia progression to dysplasia had the most significant effect on the incremental cost-effectiveness ratio. The probability sensitivity analysis concluded that the result of the NGCS having the highest cost-effectiveness was stable. Although the application of the NGCS will require upfront screening costs, it can significantly improve the detection rate of early gastric cancer and save the consequent long-term healthcare costs. It is practicable and can be popularized in China.

**Keywords:** gastric cancer; Hp; new gastric cancer screening scoring system; cost-effectiveness analysis





## 1. Introduction

Gastric cancer (GC) is one of the most common gastrointestinal cancers, which continuously develops from the normal gastric mucosa into gastritis, atrophic gastritis (AG), intestinal metaplasia (IM), dysplasia, and cancer [1]. According to the statistics of the World Health Organization in 2020, GC is estimated to have 479,000 new cases and 374,000 deaths in China, ranking fourth in incidence and third in mortality among malignant tumors [2,3]. In 2018, the treatment cost of GC in China was 23.508 billion yuan, which was the third-highest treatment cost for malignant tumors [4]. Therefore, GC not only threatens the life and health of people, but also is a serious disease burden in China. Because the symptoms of early GC are nonspecific, typically manifesting as epigastric discomfort and dyspepsia, most cases are diagnosed at advanced stages. Even if actively treated, the 5-year survival rate is less than 30% [5], while the 5-year survival rate of early GC after treatment can reach 90% [6], making early diagnosis and treatment important. Guidelines recommend screening for people at risk of GC [7]; there are various screening methods, such as Helicobacter pylori (Hp), pepsinogen (PG), gastrin-17 (G-17), tumor markers, and endoscopy. The gold standard for the diagnosis of GC is gastroscopy and biopsy. However, due to the invasiveness and poor compliance of gastroscopy, this method is not feasible for mass screening. Therefore, it is necessary to adopt a convenient, compliant, and non-invasive method for mass screening. Hp is considered the main factor that leads to GC. Although half of the people in China are infected with Hp, most people infected have no obvious symptoms [8]. Long-term persistent Hp infection leads to gastric

mucosal atrophy. According to the Correa model, if there is no intervention treatment, the infection will continue to progress unless intervened by the Hp eradication treatment [9]. Serum PG can reflect the morphological changes in the gastric mucosa and measuring their levels can predict the occurrence of precancerous lesions and GC [10]. The serum level of gastrin (mainly G-17) can also reflect the morphology of the gastric mucosa. The NGCS is a scoring system that reflects the level of GC risk for high-risk groups in China [7]. It combines five factors to systematically evaluate Hp, PG, G-17, age, and sex. Although the effect of Hp and the NGCS on the early prevention of GC is recommended by guidelines and has been confirmed by several studies, it is not listed as a routine screening item in high-risk populations of GC. Therefore, a health economic evaluation is needed to provide a theoretical basis for screening and prevention of GC.

At present, the Markov model is the most common method to evaluate the health economic aspects and has been widely used by scholars. Large-scale GC screening programs implemented in Japan and South Korea have improved the detection rate of early GC (63.3% in Japan and 63.9% in South Korea) [11,12]. In Japan, the upper gastrointestinal series (UGI) was first used for nationwide screening [13]. However, in recent years, as a result of the UGI's slow positive rate and development of other screening methods, an increasing number of scholars have studied the cost-effectiveness of serological screening, Hp screening, and endoscopic screening. Kowada applied the Markov model to analyze and evaluate the cost-effectiveness of Hp, UGI, and endoscopic screening in countries with a high incidence of GC [14]. The study showed that the Hp screening may be more cost-effective. Saito et al. used the combined detection of serum Hp IgG antibody (HPA) and serum PG ("ABC method") to assess the risk of GC and compared it with annual gastroscopy screening [15]. This study found that the "ABC method" is a cost-effective method for GC risk screening in Japan.

In countries with a high incidence of GC, the detection rate of early GC is only 10%. Therefore, early screening and prevention of GC is particularly important. However, there is still a lack of economic evaluation of the NGCS and the difference in cost-effectiveness of the NGCS compared to the Hp screening. This study applies the Markov model to evaluate the cost-effectiveness of Hp and NGCS in areas with a high incidence of GC to provide theoretical support and guide medical decisions.

## 2. Methods

### 2.1. Establish the Markov Model

The TreeAge Pro software 2011 (TreeAge Software, Williamstown, MA, USA) was used to build the Markov models, and three decisions were established: no screening, Hp screening, and NGCS screening. The decision tree is shown in Figure 1A.

(1)  No screening.
(2)  Hp screening. The 14C urea breath test was used to screen this population. The positive screening results were treated with a regular 14-day treatment: esmprazole 20 mg bid, colloidal pectin bismuth 150 mg tid, furazolidone 100 mg bid, amoxicillin 1 g bid. Those allergic to amoxicillin were replaced with tetracycline 500 mg bid. If the Hp eradication treatment fails, the next cycle still enters the Hp-positive cycle.
(3)  NGCS screening. According to the NGCS, patients were divided into three groups: low-risk, moderate-risk, and high-risk (Tables 1 and 2). The high-risk GC group was examined using a magnifying endoscopy and monitored by gastroscopy annually. The moderate-risk GC group was examined using magnifying endoscopy and monitored by gastroscopy every two years. Gastroscopy was performed every three years in the low-risk GC group [7]. If the result was dysplasia or early GC, individuals were treated with endoscopic submucosal dissection (ESD) or surgical treatment, followed by gastroscopy monitoring every half year for one year, every year for three years [16].

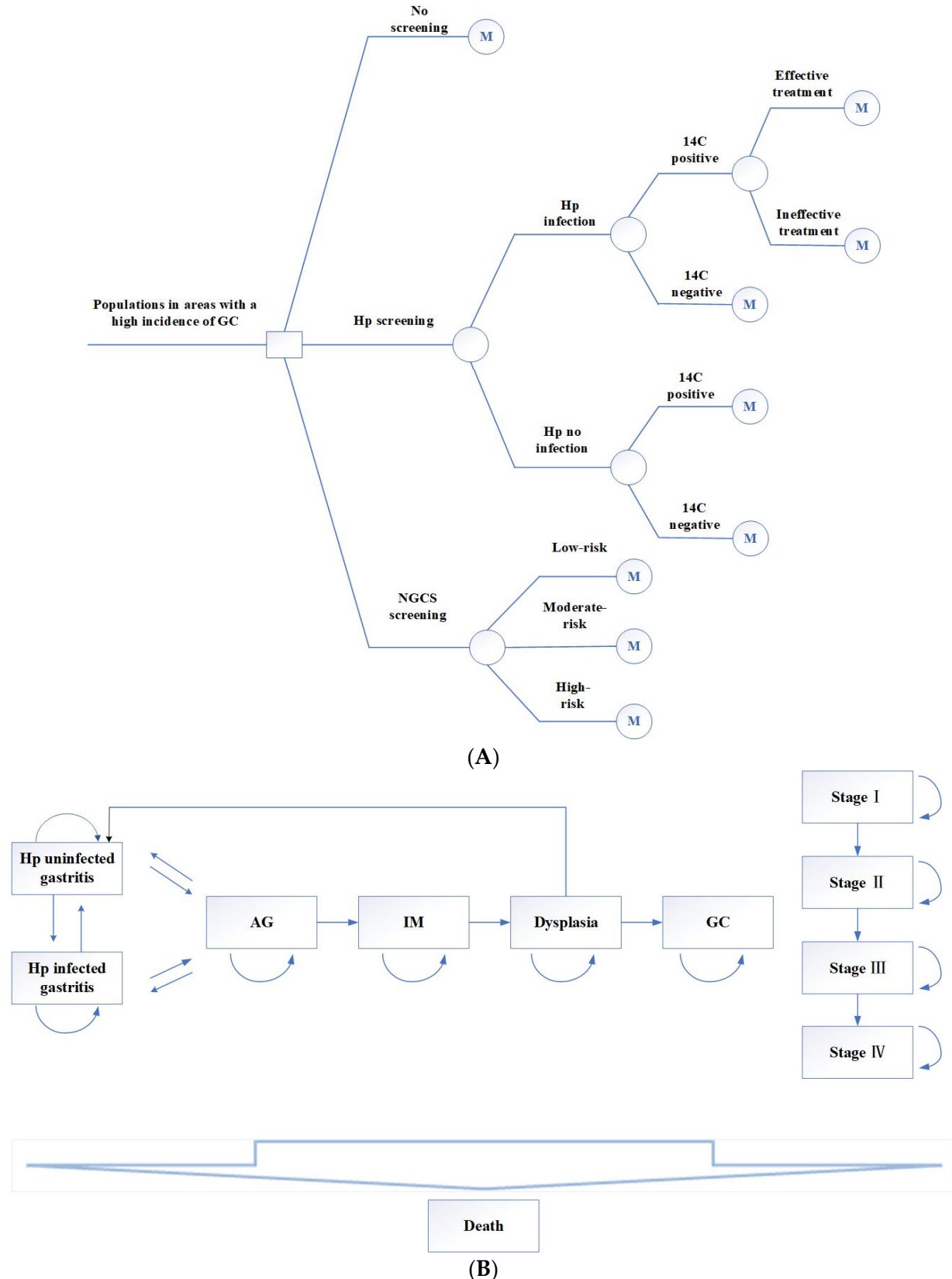

**Figure 1.** Decision tree (**A**) and Markov model (**B**) by using Hp screening and NGCS screening. Hp, Helicobacter pylori; NGCS, new gastric cancer screening scoring system; AG, atrophic gastritis; IM, intestinal metaplasia; GC, gastric cancer.

**Table 1.** New gastric cancer screening scoring system [7].

| Variables Name | Class | Score |
|---|---|---|
| Age (y) | 40~49 | 0 |
| | 50~59 | 5 |
| | 60~69 | 6 |
| | >69 | 10 |
| Sex | woman | 0 |
| | man | 4 |
| Hp infection | absent | 0 |
| | present | 1 |
| PGR | ≥3.89 | 0 |
| | <3.89 | 3 |
| G-17 (pmol/L) | <1.50 | 0 |
| | 1.50~5.70 | 3 |
| | >5.70 | 5 |

**Table 2.** Different classification standards.

| Class | Score |
|---|---|
| Low-risk | 0–11 |
| Moderate-risk | 12–16 |
| High-risk | 17–23 |

Note: Hp, Helicobacter pylori; PGR, pepsinogen ratio (PGI/PGII).

The target population of this study was a hypothetical population with a high incidence of GC aged 40 years [7]. All the participants in the Markov model entered one of the initial health states. If screening and intervention are performed, AG can be reversed to gastritis, and dysplasia can become normal after treatment. However, whether Hp eradication can reverse IM remains controversial [17–19]. In this study, Hp eradication after progression to IM could not reverse AG [17]. The model sets a cycle period of one year and takes an estimated life expectancy of 77 years as the endpoint. The Markov model is shown in Figure 1B. All parameters of this research model are from the published literature and therefore do not need ethical application.

In the model, we assumed: (1) Ignoring the compliance of screening follow-up treatment; (2) If no intervention treatment is taken, the health status will not be reversed; (3) In the no screening program, diagnosis and treatment are only accepted when clinical symptoms appear; (4) The stomach is stimulated by various foods after birth, and there will be chronic inflammation. Therefore, we assumed that the health status in our study began with the presence of gastritis.

### 2.2. Model Parameters

The parameters of the model include probability, cost, and utility parameters. The probability parameters include the prevalence of gastritis, AG, IM, dysplasia, GC stage I, II, III, IV, and the transition probability among different health states, Hp infection rate, sensitivity and specificity of 14C, Hp eradication rate and so on. The cost parameters include the treatment cost of each health state, the cost of Hp screening and NGCS screening, and the cost of Hp eradication, ESD. The utility parameters include the utility of Hp infection, various health states, and clinical diagnosis GC. The results are presented in Table S1.

### 2.3. Cost-Effectiveness Analysis (CEA)

The significance of the cost-effectiveness analysis is to evaluate whether the screening strategy is economically feasible. The cost is expressed in yuan and the effect is expressed

in quality-adjusted-life-years (QALYs). The index of the CEA adopts the incremental cost-effectiveness ratio (ICER) (Equation (1)).

$$ICER = \frac{\Delta C}{\Delta E} = \frac{C2 - C1}{E2 - E1} \tag{1}$$

where *C* is the cost and *E* is the effectiveness of each method [20].

### 2.4. Sensitivity Analysis

A one-way sensitivity analysis adjusts only one variable at a time, while the other parameters remain unchanged to obtain the ICER. The tornado diagram is used to represent the influence of various factors on the analysis results. The probability sensitivity analysis is used to evaluate all parameters through 1000 runs of the Monte Carlo simulation and obtain the scatter diagram distribution of the ICER. Further analysis yields a cost-effectiveness acceptable curve. The survival curve obtains the percentage of each health state in each cycle and connects them to form a curve.

## 3. Results

### 3.1. Cost-Effectiveness Analysis

The results from the cost-effectiveness analysis are shown in Tables 3 and 4 and Figure 2. Compared to no screening, the ICER of Hp was 536.42 yuan (Table 3) and the NGCS screening had a higher QALY (20.00893 QALY vs. 19.98036 QALY) and lower cost (50,710.36 yuan vs. 91,636.78 yuan) (Table 4). Figure 2 intuitively reflected that the Hp screening required higher costs, and the ICER was 38,680.31 yuan (Table 4), compared to the NGCS screening, and the cost of no screening is higher, and the effect is lowest in the three methods. Therefore, the application of the NGCS screening is the most cost-effective option, while no screening is the least desirable one (Figure 2).

**Table 3.** Cost-effectiveness analysis of Hp screening compared with no screening.

| Scheme | Cost (yuan) | Incremental Cost | Effect (QALY) | Incremental Effect | ICER (yuan/QALY) |
|---|---|---|---|---|---|
| No screening | 91,636.78 | - | 19.98036 | - | - |
| Hp screening | 92,227.87 | 591.09 | 21.08228 | 1.10192 | 536.42 |

Note: Hp, Helicobacter pylori; QALY, quality-adjusted life-years; ICER, incremental cost-effectiveness ratio.

**Table 4.** Cost-effectiveness analysis of NGCS and Hp screening compared with no screening.

| Scheme | Cost (yuan) | Incremental Cost | Effect (QALY) | Incremental Effect | ICER (yuan/QALY) |
|---|---|---|---|---|---|
| NGCS screening | 50,710.36 | - | 20.00893 | - | - |
| no screening | 91,636.78 | 40,926.42 | 19.98036 | −0.02857 | Dominated |
| Hp screening | 92,227.87 | 41,517.51 | 21.08228 | 1.07335 | 38,680.31 |

Note: Dominated: Compared with other methods, the cost is high, and the effect is low. NGCS, new gastric cancer screening scoring system; Hp, Helicobacter pylori; QALY, quality-adjusted life-years; ICER, incremental cost-effectiveness ratio.

### 3.2. Sensitivity Analysis

Seven parameters affected the cost-effectiveness of the NGCS screening (Figure 3A). The probability of transition from IM progression to dysplasia had the most obvious impact on ICER, indicating that the lower the probability of IM to dysplasia, the higher the cost-effectiveness and economy. Similarly, nine parameters affected the cost-effectiveness of the Hp screening (Figure 3B). The probability of the transition from Hp infecting the gastritis progression to AG had the most obvious impact on ICER, indicating that the higher the probability of Hp infecting gastritis to AG, the lower the cost-effectiveness and economy.

**Cost-Effectiveness Analysis**

**Figure 2.** Cost-effectiveness analysis of screening and no screening. Dominated: Compared with other methods, the cost is high, and the effect is low. Hp, Helicobacter pylori; NGCS, new gastric cancer screening scoring system; QALY, quality-adjusted life-years.

The results of the probability sensitivity analysis comparing the NGCS screening and no screening are shown in Figure 4A-1. The scatter points were distributed in the fourth quadrant. The baseline result (−40,926.42 yuan, 0.02857QALY) is within the 95% confidence interval of the Monte Carlo simulation result, indicating that the baseline result is reliable. The result of the NGCS screening being the most cost-effective is a stable option.

The results of the probability sensitivity analysis comparing the Hp screening and no screening are shown in Figure 4A-2. The scatter points were distributed in the first and fourth quadrant. The baseline result (591.09 yuan, 1.10192QALY) is within the 95% confidence interval of the Monte Carlo simulation result, indicating that the baseline result is reliable. From the social perspective, the Hp screening may increase or save the social cost. Taking the Gross Domestic Product per capita in 2021 (80,976 yuan) as the cost-effectiveness threshold, all the Monte Carlo simulation results are below this threshold. The result of the Hp screening having a higher cost-effective is stable.

The cost-effectiveness acceptability curve is shown in Figure 4B. When there is a willingness to pay is 0 yuan/QALY, the NGCS screening provides a 95% probability of cost-effectiveness, while the Hp screening provides only 2% probability of cost-effectiveness. When there is a willingness to pay is 80,976 yuan/QALY, the Hp screening provides a 99.4% probability of cost-effectiveness. When there is a willingness to pay is greater than 16,195.2 yuan/QALY, no screening provides a 0% probability of cost-effectiveness. Therefore, the implementation of the NGCS screening has an economic value, practicability, and can be popularized in China.

After substituting various parameters into the model for 37 cycles, the probabilities and survival curves of various health states without screening and screening using the new NGCS screening were obtained (Table 5, Figure 5). Compared to no screening, the proportions of stage III and IV were significantly reduced with the NGCS screening, which means GC can be diagnosed and treated at an early stage, the proportions of IM and

dysplasia were reduced, and the proportions of gastritis significantly increased with the use of the NGCS screening.

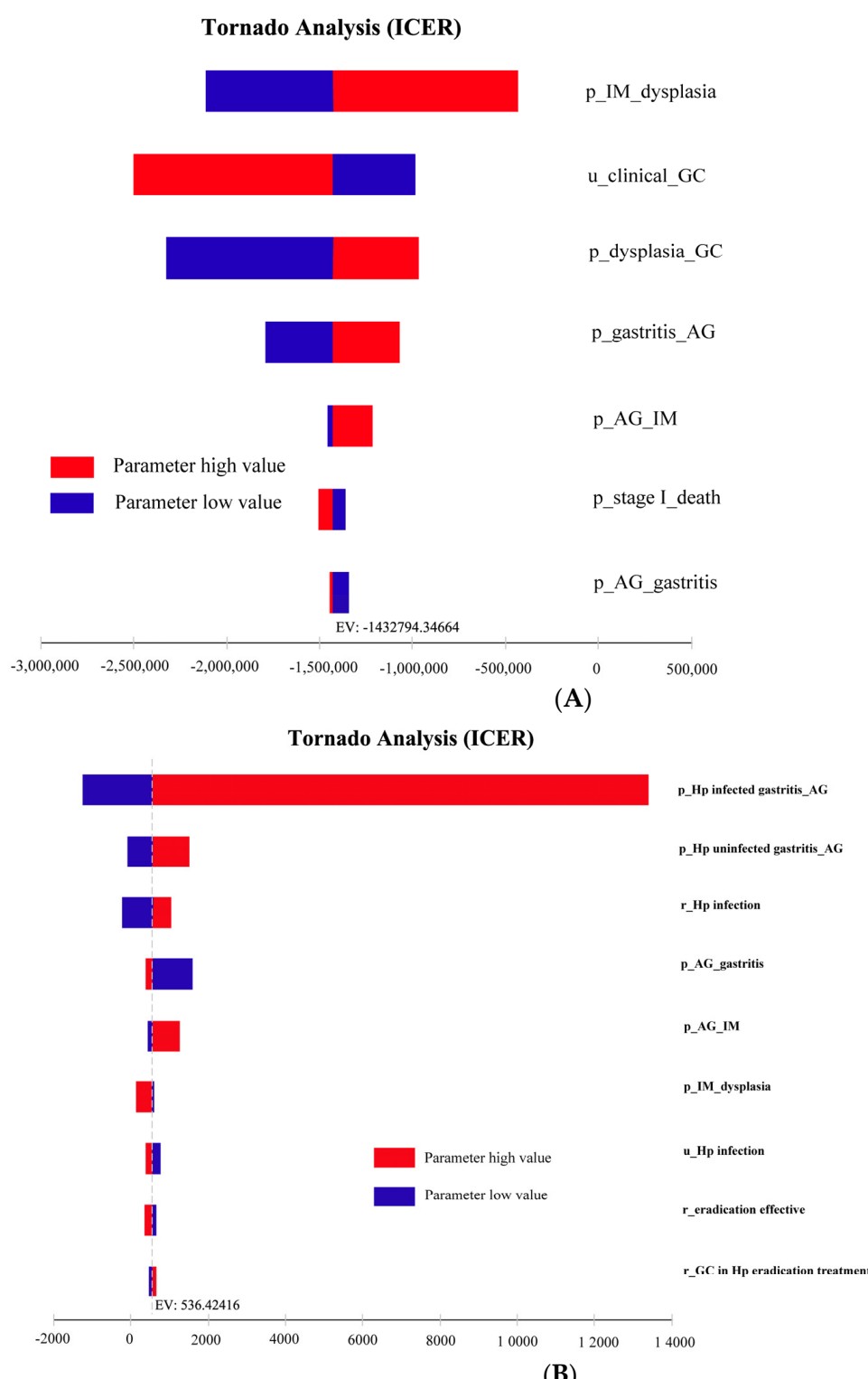

**Figure 3.** One-way sensitivity analysis of NGCS screening (**A**) and Hp screening (**B**). NGCS, new gastric cancer screening scoring system; ICER, incremental cost-effectiveness ratio; Hp, Helicobacter pylori; AG, atrophic gastritis; IM, intestinal metaplasia; GC, gastric cancer; p, probability; u, utility; r, rate.

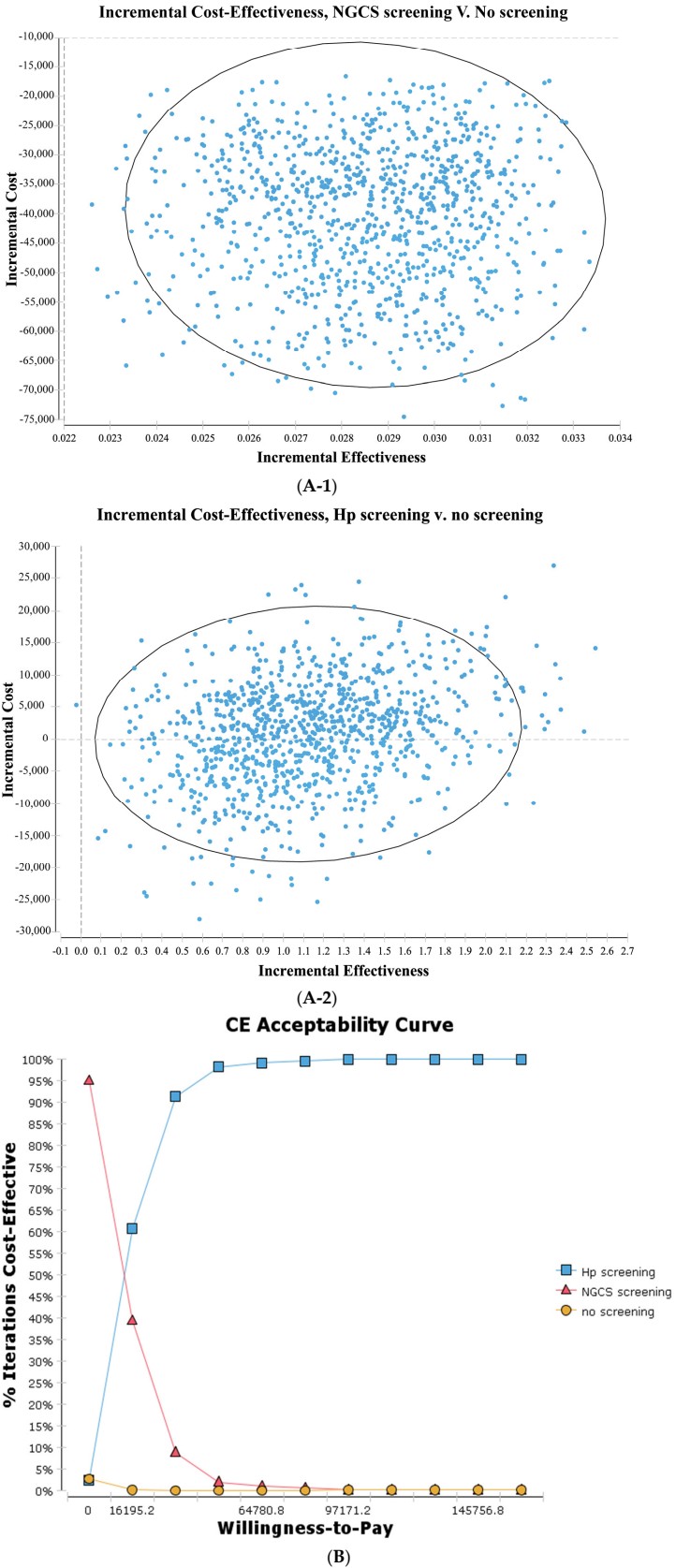

**Figure 4.** Probability sensitivity analysis (**A**) and cost-effectiveness acceptability curve (**B**) by comparing screening and no screening. (**A-1**) NGCS screening (**A-2**) Hp screening. NGCS, new gastric cancer screening scoring system; Hp, Helicobacter pylori; CE, cost-effectiveness.

**Table 5.** Probability changes of each health status in cycle 37 with no screening and with NGCS screening (%).

| | Gastritis | AG | IM | Dysplasia | Stage I | Stage II | Stage III | Stage IV | Death |
|---|---|---|---|---|---|---|---|---|---|
| no screening | 10.95 | 26.27 | 27.79 | 10.93 | 0.03 | 0.04 | 0.03 | 0.06 | 23.90 |
| Low-risk | 33.11 | 32.04 | 11.329 | 0.08 | 0.03 | 0.009 | 0.001 | 0.001 | 23.40 |
| moderate-risk | 30.17 | 31.9 | 13.85 | 0.10 | 0.14 | 0.04 | 0.005 | 0.005 | 23.79 |
| High-risk | 22.49 | 29.99 | 21.33 | 0.16 | 0.56 | 0.16 | 0.02 | 0.02 | 25.27 |

Notes: AG, atrophic gastritis; IM, intestinal metaplasia; NGCS, new gastric cancer screening scoring system.

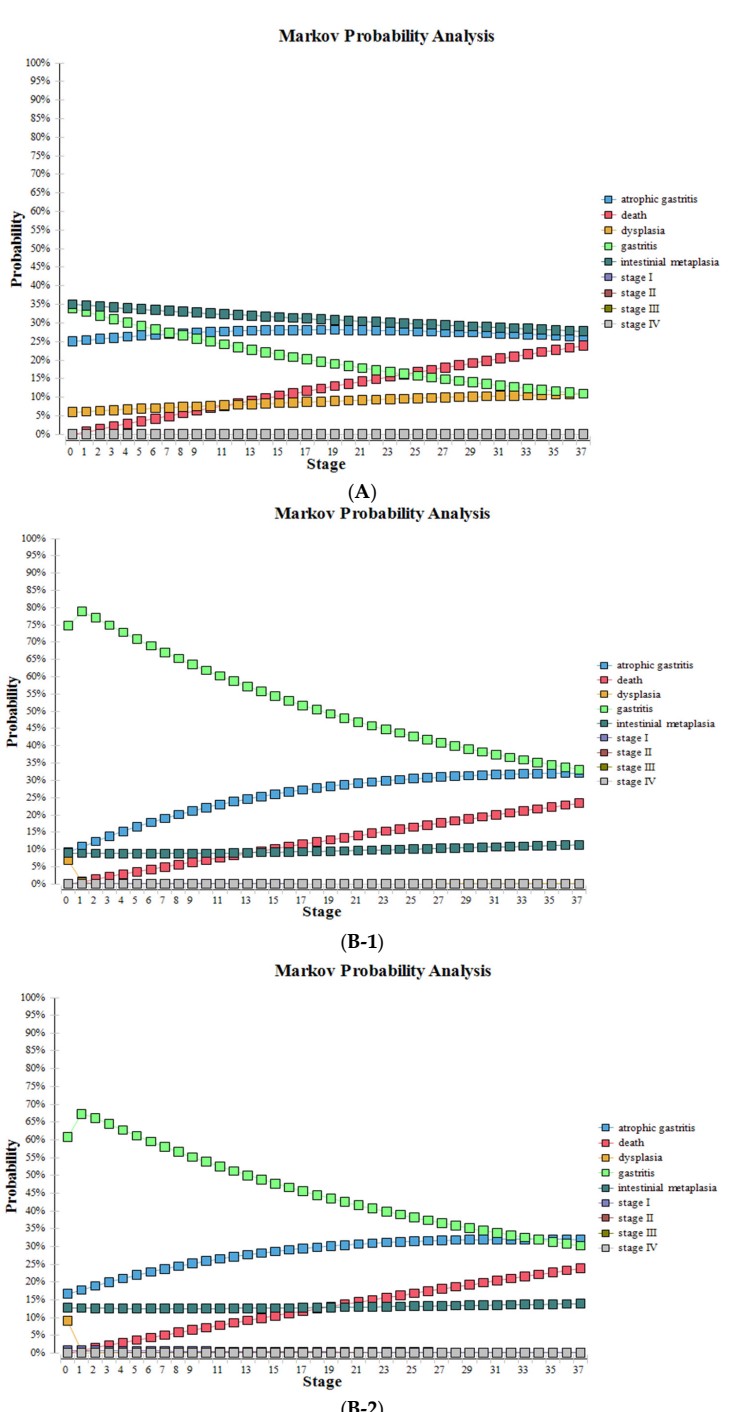

**Figure 5.** *Cont.*

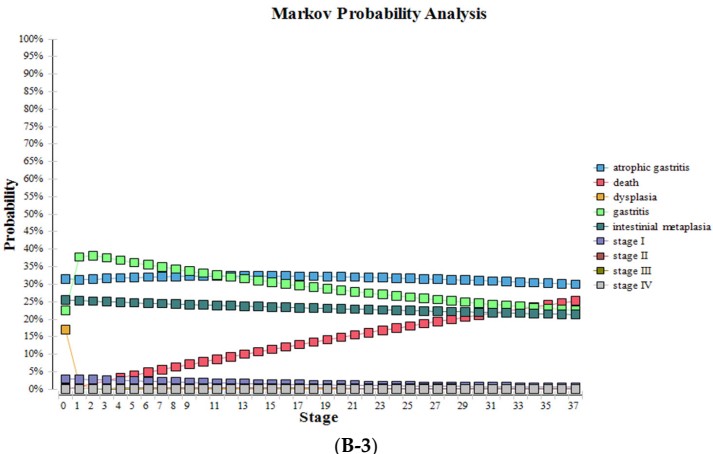

(B-3)

**Figure 5.** Health status changes by using NGCS screening and no screening. (**A**) no screening, (**B**) NGCS screening (**B-1**) low-risk group (**B-2**) moderate-risk group (**B-3**) high-risk group. NGCS, new gastric cancer screening scoring system.

## 4. Discussion

Hp is well recognized as a class I carcinogen for GC [3,8]. The urea breathing test is the most widely used non-invasive test. Eradication of Hp infection can improve the gastric mucosal inflammatory response and prevent or delay the progression of atrophy or intestinal metaplasia [8]. Compared to no screening, screening and eradication of Hp in areas with a high incidence of GC are feasible and potentially cost-effective strategies that can reduce the incidence and mortality of GC. The gold standard for the diagnosis of GC is endoscopy and biopsy. The NGCS screening combined Hp detection and eradication with endoscopic screening, which obtained additional benefits from endoscopic examination than the Hp screening alone.

This study found that the NGCS screening can reduce costs and increase QALYs compared to no screening. Although it requires investments into screening, it can significantly reduce the long-term costs of GC. This study demonstrated that the NGCS screening has the highest cost-effectiveness and is a feasible solution that can be readily adopted in China compared to the Hp screening and no screening.

At present, there are no reports on the cost-effectiveness of the NGCS screening for the screening and prevention of GC. However, the cost-effectiveness of the Hp screening has been studied by several scholars. Kowada assessed the cost-effectiveness of HPA screening and treatment in high-risk groups in Japan and concluded that HPA screening was more cost-effective than no screening in any age group. Under the willingness to pay a threshold of $10,0000/QALY, the value of the screening was 100% [21]. Saito believes that the "ABC method" is preferable to annual gastroscopy screening [15]. Yeh evaluated the cost-effectiveness of PG screening, gastroscopy screening, and Hp screening, and found that the PG screening was more cost-effective than the gastroscopy screening and Hp screening [22]. This study is consistent with the above research methods, suggesting that screening in high-risk groups for GC is cost-effective. Although no study has compared the Hp screening with the new NGCS screening, Saito and Yeh showed that the application of serological screening is the most cost-effective, consistent with the conclusion of this study.

The results of the one-way sensitivity analysis of the model showed that the probability of IM to dysplasia had the most obvious effect. Saumoy et al. found that the probability of transition from IM to dysplasia is sensitive to the model, similar to our results [23], suggesting that keen attention should be made to the key factors affecting the results when implementing a screening method, and early identification of IM can improve the overall cost-effectiveness.

The probability sensitivity analysis showed that the NGCS screening was the most cost-effective and stable. The cost-effectiveness acceptability curve showed that the NGCS screening is cost-effective without the initial investment. Saito et al. conducted a probability

sensitivity analysis of the screening and assessment of GC risk using the "ABC method" and concluded that the "ABC method" is cost-effective and stable. Further cost-effectiveness acceptability curves show that when the willingness to pay a threshold of $10,000 USD/QALY, the probability of the "ABC method" being cost-effective is 99.7% [15]. This finding is similar to that of the present study. With the improvement of China's national economic status and increasing population concern for individual health, although the application of the NGCS screening requires investment into screening costs, this screening method can significantly reduce the costs related to GC through early detection and treatment. At present, GC screening in China is limited in few areas such as near the Huaihe River, without large-scale screening. In this study, people in areas with a high incidence of GC were divided into three groups by using the NGCS and, if appropriate surveillance was implemented, it would improve compliance and save health resources. Therefore, the applicability of a NGCS screening for the high-risk population of GC in China is practical and can be popularized in the population.

This is the first study in China to evaluate the cost-effectiveness of using the new NGCS to screen and prevent GC in groups at high-risk of GC. At present, the efficacy of the NGCS screening has been confirmed by various studies and is recommended by guidelines; however, no economic evaluation of its long-term outcome has been conducted as yet. This study provides a solid theoretical basis for NGCS's implementation to potentially reduce the high cost and low survival rate of advanced GC. Then, the Markov model constructed in this study simulates the progression of GC. The health state based on the recognized developmental state of GC (Correa model) and is applicable in clinical practice. Thus, the feasibility is stronger and the recommended method of cost-effectiveness analysis in its state is easier to implement.

This study has some limitations. Firstly, the 14C urea breath test was used to screen for Hp, but other diagnostic methods for Hp were not considered. The cost-effectiveness of various methods for diagnosing Hp can be compared further. Secondly, the Markov model used in this study follows the Correa model for "intestinal" type GC but has no interventional effect on "gastric" type GC. Thirdly, we did not consider other risk factors for GC, such as long-term smoking, dietary habits, and other environmental factors. Moreover, we used perfect compliance to these two screening methods. However, this assumption provided the model with the ability to predict the maximum achievable benefits of public health strategies. Finally, the generalizability of this study is limited, as the target population was restricted to those at high-risk for gastric cancer in China. The Hp infection rate, incidence rate of GC, and medical costs vary in different countries. Therefore, these results cannot be extended to all countries, and further cost-effectiveness studies on each country's differences are needed. However, this study provides a reference for cost-effectiveness analysis of screening programs in various countries.

China has a high incidence of GC. Large-scale screening can increase the detection rate of early GC and improve survival rates. If all individuals undergo endoscopic screening, endoscopy resources will not be sufficient to meet the huge demand because of the greater population in China. NGCS screening, as the most cost-effective method, is practical and suitable for large-scale screening, and the cost is much less than per capita GDP, and this cost may be offset by the reduction in the GC disease burden. The population is divided into three risk levels according to the NGCS, and each level is reasonably monitored, which improves compliance with endoscopy. Therefore, health resources can be allocated more reasonably, maximizing the social and economic benefits of GC screening [24].

## 5. Conclusions

The new gastric cancer screening scoring system is cost-effectiveness and suitable for large-scale screening in China. Early diagnosis and treatment of gastric cancer is necessary for large countries with gastric cancer. This study provides useful insights and important ideas and directions for clinical decision-making.

**Supplementary Materials:** The following supporting information can be downloaded at: https://www.mdpi.com/article/10.3390/curroncol30010086/s1, Table S1: Parameters of the Markov model [8,11,15,21,25–56].

**Author Contributions:** Conceptualization, P.Z.; methodology, P.Z. and J.L.; software, P.Z.; validation, P.Z. and J.L.; formal analysis, P.Z. and J.L.; data curation, P.Z.; writing—original draft preparation, P.Z. and J.L.; writing—review and editing, P.Z. and J.L.; visualization, P.Z. and J.L.; supervision, J.L.; project administration, J.L.; funding acquisition, J.L. All authors have read and agreed to the published version of the manuscript.

**Funding:** This research was funded by the Scientific and Technological Achievements Transformation Guidance Special of Shanxi Province, China, grant number [201904D131030].

**Institutional Review Board Statement:** Not applicable.

**Informed Consent Statement:** Not applicable.

**Data Availability Statement:** The data presented in this study are available on request from the corresponding author.

**Acknowledgments:** The authors thank the Scientific and Technological Achievements Transformation Guidance Special of Shanxi Province, China (201904D131030) for financial support of this study.

**Conflicts of Interest:** The authors declare no conflict of interest.

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
