# Peer review of "Cost-Effectiveness Analysis of Hp and New Gastric Cancer Screening Scoring System for Screening and Prevention of Gastric Cancer"

_curroncol, doi:10.3390/curroncol30010086_

Round 1
Reviewer 1 Report
Thank you for giving me the opportunity to review this manuscript. The paper is well written and the concepts are clearly described. I have just few minor comments that I think the authors could consider for further improvement of the paper:
1. In section 2.3, CEA analysis it would be good to present the formula for ICER.
2. Please describe Figure 2 with more details. The cost effectiveness plane should actually describe the three treatments, which is more dominant and cost effective according to the relevant plane/quadrants. Only then, it will be much more easier to visualize and subsequently appropriate to discuss on the most likely treatment to be adopted by health experts for practice in the discussion part.
3. Figure 4(A) - there seem to be a typo at the title "effectiveness"
4. The discussion could be further enhanced by discussing the implications for practice by the practitioners and stakeholder likely perspectives for the adoption of the most likely treatment.
Author Response
Open Review
Comments and Suggestions for Authors
Thank you for giving me the opportunity to review this manuscript. The paper is well written and the concepts are clearly described. I have just few minor comments that I think the authors could consider for further improvement of the paper:
- In section 2.3, CEA analysis it would be good to present the formula for ICER.
Response: Thanks for your advice, we modified this in the manuscript. See line 136-137.
(1) Where C is the cost and E is the effectiveness of each method [20].
- Please describe Figure 2 with more details. The cost effectiveness plane should actually describe the three treatments, which is more dominant and cost effective according to the relevant plane/quadrants. Only then, it will be much more easier to visualize and subsequently appropriate to discuss on the most likely treatment to be adopted by health experts for practice in the discussion part.
Response: Thanks for your advice, we modified this in the manuscript. See line 151-154,
Figure 2 intuitively reflected that Hp screening required higher costs, and the ICER was 38680.31 yuan (Table 3), compared to the NGCS screening, and the cost of no screening is higher, and the effect is lowest in the three methods.
- Figure 4(A) - there seem to be a typo at the title "effectiveness"
Response: Thanks for your advice, we modified this in the manuscript. See line 206-207.
- The discussion could be further enhanced by discussing the implications for practice by the practitioners and stakeholder likely perspectives for the adoption of the most likely treatment.
Response: Thanks for your advice, we modified this in the manuscript. See line 319-329.
China has a high incidence of GC. Large-scale screening can increase the detection rate of early GC and improve survival rates. If all individuals undergo endoscopic screening, endoscopy resources will not be sufficient to meet the huge demand because of the greater population in China. NGCS screening, as the most cost-effective method, is practical and suitable for large-scale screening, and the cost is much less than per capita GDP, and this cost may be offset by the reduction of the GC disease burden. The population is divided into three risk levels according to the NGCS, and each level is reasonably monitored, which improves compliance with endoscopy. Therefore, health resources can be allocated more reasonably, maximizing the social and economic benefits of GC screening [56].

Reviewer 2 Report
In this manuscript, the authors aim to evaluate the cost-effectiveness of various modalities of gastric cancer (GC) screening, including the new gastric cancer screening scoring system(NGCS), through a decision-analytic Markov model. They show that NGCS has the highest cost-effectiveness, with consistent results in the sensitivity analysis.
However, there are different issues that remain to be addressed by the authors, as follows:
1- Hp screening appears to have a similar cost to no screening (and higher than that of NGCS screening), but a substantial benefit in relation to both no screening and NGCS screening in terms of incremental effect in QALY. This is represented by a low ICER when comapring Hp screening to no screening, and a high ICER when compared to NGCS screening. Consequently, Hp screening merits further consideration. However, the authors discuss minimally the value of Hp screening when presenting the results and do not include this modality in most the following analyses performed, including the sensitivity analysis.
2- While this study is interesting in that it attempts to perform a simulation of the cost-effectiveness of various modalities for GC screening as compared to no screening, one of its major limitations is that it ignores (as part of the assumptions detailed in the simulation performed) many relevant parameters of real-world practice, including compliance. Considering the limited benefit in terms of QALYs of NGCS screening (as compared to no screening), the influence of such non-considered parameters could be major.
3- In the cost-effectiveness acceptability curve (shown in Figure 4b), the authors evaluate only two options (i.e. no screening and NGCS screening). As Hp screening yields higher QALYs, it would be important to evaluate this third option in the CE acceptability curve.
4- In Figure 1A: what does M mean in terms of abbreviations in the different circles at the end of tree branches?
Author Response
Comments and Suggestions for Authors
Line30 Need to elaborate more on magnitude of the problem (epidemiologically and economically) in china
Response: Thanks for your advice, we modified this in the manuscript. See line 29-33.
According to the statistics of the World Health Organization in 2020, GC is estimated to be 479, 000 new cases and 374, 000 deaths in China, ranking fourth in incidence and third in mortality among malignant tumors [2,3]. In 2018, the treatment cost of GC in China was 23.508 billion yuan, which was the third-highest treatment cost for malig-nant tumors [4].
Line 76 Add developing company and version if available
Response: Thanks for your advice, we modified this in the manuscript. See line 80-81.
The TreeAge Pro software 2011 (TreeAge Software, Williamstown, Massachusetts, USA)
Line 79 what are population characteristic for no screening
Response: Thanks for your advice, we used some sentences to explain this question. (The target population of this model study was a hypothetical population with a high incidence of GC aged 40 years. It established three decisions and no screening was used as the control group for comparison.)
Line 80 Add “urea breath test”
Response: Thanks for your advice, we modified this in the manuscript. See line 84.
Line 83 Missing: if Hp eradication treatment fail
Response: Thanks for your advice, we modified this in the manuscript. See line 87-88.
Line 96-97 Needs a reference
Response: Thanks for your advice, we modified this in the manuscript. See line 101.
Line 125-126 table1 caption is different from what is mentioned in the paragraph
Response: Thanks for your advice. The model parameters are presented in Table s1 and Table 1 is New gastric cancer screening scoring system (NGCS).
Line 151 Replace with NGCS
Response: Thanks for your advice, we modified this in the manuscript. See line 159.

Reviewer 3 Report
Greetings Authors
Kindly comply with the amendments that appear in the attached manuscript file.
Thank you.

Author Response
Comments and Suggestions for Authors
In this manuscript, the authors aim to evaluate the cost-effectiveness of various modalities of gastric cancer (GC) screening, including the new gastric cancer screening scoring system(NGCS), through a decision-analytic Markov model. They show that NGCS has the highest cost-effectiveness, with consistent results in the sensitivity analysis.
However, there are different issues that remain to be addressed by the authors, as follows:
- Hp screening appears to have a similar cost to no screening (and higher than that of NGCS screening), but a substantial benefit in relation to both no screening and NGCS screening in terms of incremental effect in QALY. This is represented by a low ICER when comapring Hp screening to no screening, and a high ICER when compared to NGCS screening. Consequently, Hp screening merits further consideration. However, the authors discuss minimally the value of Hp screening when presenting the results and do not include this modality in most the following analyses performed, including the sensitivity analysis.
Response: Thanks for your advice.
The main purpose of this study is to evaluate Hp screening and NGCS screening, which is the most cost-effective option, and no screening as the control group.After operation model by TreeagePro software,it is concluded that NGCS screening is the most cost-effective , so the NGCS screening is emphatically discussed.
we modified this in the manuscript. See line 243-251.
Hp is well recognized as a class I carcinogen for GC [3,8]. Urea breathing test is the most widely used non-invasive test. Eradication of Hp infection can improve the gastric mucosal inflammatory response, and prevent or delay the progression of atrophy or intestinal metaplasia[8]. Compared to no screening, screening and eradication of Hp in areas with a high incidence of GC is a feasible and potentially cost-effective strategy that can reduce the the incidence and mortality of GC. The gold standard for the diagnosis of GC is endoscopy and biopsy. The NGCS screening combined Hp detection and eradication with endoscopic screening, which obtained additional benefits from endoscopic examination than Hp screening alone.
See line 171-175.
Similarly, nine parameters affected the cost- effectiveness of the Hp screening (Figure 3B). The probability of transition from Hp infected gastritis progression to AG had the most obvious impact on ICER, indicating that the higher the probability of Hp infected gastritis to AG, the lower the cost-effectiveness and economy.
See Figure 3B
See line 190-197.
The results of the probability sensitivity analysis comparing the Hp screening and no screening are shown in Figure 4Aâ‘¡. The scatter points were distributed in the first and fourth quadrant. The baseline result (591.09 yuan, 1.10192QALY) is within the 95% confidence interval of the Monte Carlo simulation result, indicating that the baseline result is reliable. From the social perspective, Hp screening may increase or save the social cost. Taking the Gross Domestic Product per capita in 2021(80976 yuan) as the cost-effectiveness threshold, all the Monte Carlo simulation results are below this threshold, The result of Hp screening has a higher cost-effective is stable.
See Figure 4 A â‘¡
- While this study is interesting in that it attempts to perform a simulation of the cost-effectiveness of various modalities for GC screening as compared to no screening, one of its major limitations is that it ignores (as part of the assumptions detailed in the simulation performed) many relevant parameters of real-world practice, including compliance. Considering the limited benefit in terms of QALYs of NGCS screening (as compared to no screening), the influence of such non-considered parameters could be major.
Response: Thanks for your advice.
we used perfect compliance to these two screening methods, because the compliance of patients is difficult to assess. we should improve patient’s compliance as much as possible during screening. First of all, screening in areas with a high incidence of gastric cancer is funded by the goverment, so the poor compliance caused by funds can be ignored. Secondly, screening in areas with a high incidence of gastric cancer to improve the detection rate of early gastric cancer is recommended by the guidelines and is extremely necessary. Health education should be carried out to improve people's awareness of the disease, Thirdly, the screening is concentrated in one area, family and community supervision and support can greatly improve compliance. Finally, medical workers should keep patience and maintain a harmonious doctor-patient relationship.
we modified this in the manuscript. See line 310-312.
Then, we used perfect compliance to these two screening methods. However, this as-sumption provided the model with the ability to predict the maximum achievable benefits of public health strategies.
3- In the cost-effectiveness acceptability curve (shown in Figure 4b), the authors evaluate only two options (i.e. no screening and NGCS screening). As Hp screening yields higher QALYs, it would be important to evaluate this third option in the CE acceptability curve.
Response: Thanks for your advice, we modified this in the manuscript. See line 198-204 and Figure 4B.
When willingness to pay is 0 yuan / QALY, the NGCS screening provides a 95% proba-bility of cost-effectiveness, while the Hp screening provides only 2% probability of cost-effectiveness; When willingness to pay is 80976 yuan / QALY, the Hp screening provides a 99.4% probability of cost-effectiveness. When willingness to pay is greater than 16195.2yuan / QALY, the no screening provides a 0% probability of cost-effectiveness.
4- In Figure 1A: what does M mean in terms of abbreviations in the different circles at the end of tree branches?
Response: Thanks for your advice, M means that each branch enters the Markov mode. All the participants in the Markov model entered one of the initial health states.

Round 2
Reviewer 1 Report
Thank you for revising the manuscript based on my previous suggestions.
Reviewer 3 Report
The authors replied to all comments and amendments and the manuscript in its revised version ready for publication.